# Vitamin D and Microbiota: Is There a Link with Allergies?

**DOI:** 10.3390/ijms22084288

**Published:** 2021-04-20

**Authors:** Giuseppe Murdaca, Alessandra Gerosa, Francesca Paladin, Lorena Petrocchi, Sara Banchero, Sebastiano Gangemi

**Affiliations:** 1Department of Internal Medicine, University of Genoa, 16132 Genoa, Italy; a.gerosa92@gmail.com (A.G.); puell-a@hotmail.it (F.P.); lory.petra@gmail.com (L.P.); sarabanchero@hotmail.it (S.B.); 2Ospedale Policlinico San Martino IRCCS, 16132 Genoa, Italy; 3School and Operative Unit of Allergy and Clinical Immunology, Department of Clinical and Experimental Medicine, University of Messina, 98125 Messina, Italy; gangemis@unime.it

**Keywords:** vitamin D, microbiota, immune system, allergies

## Abstract

There is increasing recognition of the importance of both the microbiome and vitamin D in states of health and disease. Microbiome studies have already demonstrated unique microbial patterns in systemic autoimmune diseases such as inflammatory bowel disease, rheumatoid arthritis, and systemic lupus erythematosus. Dysbiosis also seems to be associated with allergies, in particular asthma, atopic dermatitis, and food allergy. Even though the effect of vitamin D supplementation on these pathologies is still unknown, vitamin D deficiency deeply influences the microbiome by altering the microbiome composition and the integrity of the gut epithelial barrier. It also influences the immune system mainly through the vitamin D receptor (VDR). In this review, we summarize the influence of the microbiome and vitamin D on the immune system with a particular focus on allergic diseases and we discuss the necessity of further studies on the use of probiotics and of a correct intake of vitamin D.

## 1. Introduction

In the last decade, a lot of studies have been conducted on the effects of both the microbiome and vitamin D on the immune system. Research indicates that the microbiota exhibits both pro-inflammatory and anti-inflammatory properties in direct or indirect ways [1]. It is now clear that alterations of the normal composition of the microbiome are narrowly associated with immunological disorders [2]. Vitamin D is a fat-soluble vitamin and a critical regulator of calcium and phosphate homeostasis and bone health [3]. Among other systemic effects, vitamin D, mainly through the vitamin D receptor (VDR), also has an important role in the modulation of immune response [4,5]. Vitamin D and the microbiome seem to affect the immune system in various yet similar ways, but is this due to an interaction and/or a synergistic effect between them? As a matter of fact, interactions between vitamin D, gut bacteria, and the immune system can occur at several levels and may include both the innate and the adaptive immune system. The aim of this review is to underline the relationship between the microbiome, vitamin D, and the immune system with particular attention on allergic reactions.

## 2. Microbiome and Microbiota

The human microbiota is the community of commensal, symbiotic, and pathogenic microorganisms that survive on our body, skin, and respiratory, gastrointestinal and urogenital systems [6]. The composition of the microbiota is already formed in the early years of life but is dynamic and shaped by both genetic and non-genetic factors [2]. The microbiome is the set of genomes of our microbial symbionts [7]. Thanks to the advancement of technology, the genes of the microbial communities that colonize our organism have been sequenced over the last 10 years. The largest datasets reported come from the gut. The human microbiota consists of 12 different bacterial phyla, with 93.5% classified as *Bacteroidetes, Proteobacteria, Firmicutes, Actinobacteria, or Euryarchaeota phyla* [6]. Pathogenic microorganisms can be distinguished from commensal microbiota because they carry specific adhesive and invasive molecules, such as adhesins and invasine, which allow them to adhere and invade tissues, causing damage to the host. Early microbial colonization of the mucous membranes, such as the respiratory system and skin, occurs in tandem with the development of the immune system. During early microbial assembly, the immune system is susceptible to the colonization of organisms due to its immaturity: The decreased secretion of cytokines results in muted inflammatory responses, which allows the settlement and expansion of the microbiome in the various niches [8]. Germ-free mice provided key evidence on the importance of the microbiota to health, as these animals showed various immune defects and increased susceptibility to infections [1].

Figure 1 describes the regulatory effects of microbial exposure on Th2 cells.

### 2.1. Intestinal Microbiota

In recent years it has become evident that the gut microbiome plays a vital role in shaping the immune system and contributing to the pathogenesis of many diseases. It was found to be essential for host immune function, nutrient digestion, the production of short-chain fatty acids (SCFA), vitamin synthesis, energy metabolism, intestinal permeability, protection from pathogens, and the determination of susceptibility of the host to gastrointestinal infections [9]. The colonization of a newborn’s gut can begin before, during, or after delivery. In fact, bacteria have been detected in umbilical cord blood and amniotic fluid, proving that the fetal gastrointestinal tract can already harbor a limited prenatal microbiome. During birth and in the immediate postnatal period, the baby is exposed to microorganisms of maternal origin (carried by the vaginal microbiota and from feces and skin) and the environment. Many of these microbes have the potential to rapidly colonize the newborn′s gut: The bacteria are cultured from newborn feces within hours of birth [10]. In addition to the maternal vaginal and intestinal flora, breast milk also helps to shape the infant microbiome, as it is heavily enriched with germs. The gut microbial community is the most abundant [2] and the vast majority of microbiota in the human gastrointestinal tract live in the colon: The gut contains a large and diverse population of microorganisms that is, quantitatively, the most important postnatal source of microbial stimulation of the immune system. Anaerobes (particularly *Firmicutes* and Gram-positive *Actinobacteria* and Gram-negative *Bacteroidetes*) are the predominant bacteria in the gastrointestinal tract of adults [11]. It has been shown that the effects of microbial factors derived from the intestine are not limited to the intestinal microenvironment, but also to the immune cells of other organs, through the so-called intestine–lung, intestine–brain, and intestine–bone axes, and others. These broad effects could be mediated by the release of metabolites produced by the gut microbiota into the circulation or by the trafficking of gut-derived activated T cells to other compartments of the body [2]. As a result, the gut microbiota could affect the whole organism′s immune system.

### 2.2. Airway Microbiota

The previously accepted idea of “sterile lungs” was challenged by the discovery of the airway microbiota, demonstrated by the introduction of lung tissues in the Human Microbiome Project. Distinct airway microbial patterns begin to form immediately after birth. It has been hypothesized that the lower airway microbiome is derived from that of the upper airways via microbial aspiration or direct inhalation, to a lesser extent. In healthy subjects, the respiratory microbiome has a low density and a modest growth rate [7]. Even though the airway microbiota is less studied than that of other areas, it is know that the composition of the respiratory system microbiome differs in health and disease conditions. In state of health, the commensal bacteria are beneficial for humans by strengthening the immune system. On the contrary, if the microbiome is disturbed, for example because of an antibiotic therapy, potential pathogens may proliferate locally or be transmitted to other areas, causing respiratory or systemic infections, allergies, and asthma [12].

### 2.3. Skin Microbiota

The term “skin microbiota” refers to the microorganisms that reside on human skin. In most cases these are bacteria: There are in fact about 1000 species on human skin, categorized into 19 phyla [13]. The majority of the skin microbiota is found in the superficial layers of the epidermis and in the upper parts of the hair follicles. Skin flora is generally non-pathogenic; the resident bacteria can prevent infections from pathogenic organisms, either by competing for nutrients, secreting chemicals against them, or stimulating the skin′s immune system.

## 3. Vitamin D

### 3.1. Synthesis and Metabolism 

Vitamin D encompasses both vitamin D2 (ergocalciferol) and vitamin D3 (cholecalciferol) [14]. In humans, the major source of vitamin D (90%) is the exposure to solar UVB radiation, which determines the formation of cholecalciferol in the skin, which is then metabolized in the liver, by the vitamin D 25-hydroxylase Cyp2R1 and to a lesser extent by Cyp27A1, to 25-hydroxyvitamin D (25-OH-D3) and finally carried to the kidneys, where it is transformed into the active form (1,25-dihydroxyvitamin D, 1,25-(OH)2D) [3]. Only 10% of vitamin D is obtained through food ingestion (with vitamin D-rich foods such as cod liver oil, tuna, sardines, milk, eggs, certain mushrooms, and fortified orange juice and dairy products) [15].

### 3.2. Mechanism of Action

l,25(OH)2D3 acts primarily through vitamin D receptors (VDR) [16]. VDR is a nuclear hormone receptor and transcription factor expressed in a variety of tissues, including the intestines, adipose tissue, and liver, as well as most immune cells, and modulates metabolic and immune system processes [17]. VDR has a key role in the modulation of the immune response since it is expressed in immune cells, including CD4^+^ and CD8^+^ T cells, B cells, neutrophils, and antigen-presenting cells (APCs) [4]. Many of these cells, such as macrophages and dendritic cells, are capable of synthesizing biologically active vitamin D from circulating 25OHD, which enables the rapid increase of local levels of vitamin D, potentially needed to shape adaptive immune responses [18]. VDR is also highly present in the small intestine and colon, where it plays critical roles in proliferation, differentiation, permeability, host–microbial interactions, immunity, and susceptibility to pathogenic infection. Notably, it is crucial for maintaining a healthy microbiome [19].

Figure 2 describes the effects of the VD/VDR complex on the immune system.

### 3.3. Systemic Effects and Deficiency

Vitamin D is an important regulator of the immune system, acting directly on immune cells to promote an anti-inflammatory state, and the balance between proinflammatory and anti-inflammatory activity is disrupted in vitamin D deficiency in favor of the former [6]. Populations with lower levels of vitamin D (i.e., those living furthest from the equator and those in early infancy) are more likely to develop several immune-mediated diseases, including allergic asthma and allergies to foods [14,20]. There is solid evidence that vitamin D supplementation can reduce the rate of infection, preventing autoimmune disorders, and there is promising data linking vitamin D deficiency to increased rates of childhood asthma and other allergic conditions [21]. Despite this, what represents adequate levels of vitamin D in the blood for human health generally and specifically for each of the reviewed allergic conditions remains controversial, as some observational studies seemed to confirm that vitamin D deficiency may contribute to increasing the risk of allergy and asthma. The necessity for further studies in this field is evident [18,22].

## 4. Linking Vitamin D, the Microbiome, and the Immune System

Both vitamin D deficiency and dysbiosis have been shown to impact systemic and chronic inflammation and to increase the risk of various conditions, including cardiovascular, neurological, infectious (including COVID-19), and metabolic diseases, autoimmune disorders, and cancer [4,5,23,24]. Focusing on allergies, their pathogenesis can be explained by both the hygiene hypothesis and the vitamin D hypothesis. The hygiene hypothesis was first proposed in 1989 by David P. Strachan [25], who postulated that infections in early childhood, transmitted by unhygienic contact with older siblings or acquired prenatally from the mother, could prevent the development of allergic diseases. An evolution of this theory is the old friend hypothesis. According to Rook [26], a hygienic lifestyle and cleanliness can be defined as an abuse of antibiotics, antibacterial soaps, and cleaners; delayed exposure to viruses; and an excessive time spent indoors. All of these can decrease immune tolerance and deplete indigenous commensal bacteria (the “old friends”). According to the vitamin D hypothesis, adequate vitamin D levels and supplementation in the first year of life can sensitize children against allergens and reduce the risk of development of food allergies and asthma [27]. On the other hand, high levels of vitamin D may also increase the risk of allergic sensitization by inhibiting the maturation of dendritic cells and the development of T-helper 1 responses [28]. Research into the gut microbiome and vitamin D is therefore considered to be promising for the understanding, treatment, and prevention of autoimmune and allergic diseases [29].

The effects of hypovitaminosis D and dysbiosis on immune system are depicted in Figure 3.

### 4.1. Innate Immune System

The innate immune response system forms an important line of defense. It involves cells of hematopoietic origin (including macrophages, mast cells, neutrophils, eosinophils, dendritic cells, and natural killer cells) and nonhematopoietic components (skin and epithelial cells lining the gastrointestinal, genitourinary, and respiratory tracts). The cellular defenses are further supplemented by humoral components, which include complement proteins, C-reactive protein, and lipopolysaccharide (LPS)-binding protein [30]. Innate immunity is finely regulated in the gut, and innate immune cell subsets have been identified in both murine and human intestinal lamina propria [31]. In the last 10 years multiple levels of interaction between the microbiota and the cells of the innate immune system have been uncovered. Intestinal epithelial cells present an extensive repertoire of innate immune receptors (PRRs, TLRs, and the NOD-like receptors NOD1 and NOD2) that are essential for maintaining the integrity of the intestinal barrier and the production of antimicrobial peptides [32]. Once the epithelial layer is compromised and bacteria or their antigenic products enter the lamina propria, a rapid immune response is activated by resident sentinel cells of the innate immune system. In the quiescent M2 state, resident macrophages respond to the invasion of the bacteria by activating the NFκB pathway [33]. Among the extra-skeletal effects of vitamin D, a particularly important one is exhibited in innate immunity. In particular, vitamin D stimulates the production of pattern recognition receptors, anti-microbial peptides, and cytokines [34]. In addition to this, intestinal epithelial VDR regulates autophagy and innate immune functions through the autophagy gene ATG16L1, which could change the microbiota profile [35].

#### Innate Lymphoid Cells

Innate lymphoid cells (ILCs) are a family of innate immune cells that belong to the innate immune system but develop from the lymphoid lineage. Contrary to T and B lymphocytes, ILCs do not have RAG-mediated recombined antigen receptors. ILC2s are known to produce Th2 signature cytokines (IL-4, IL-13, IL-9, IL-5, and IL-6) and their main function is to promote type-2 inflammation, which is important during allergies, helminth infection, and tissue repair. ILC2s are found in different tissues, including lung and adipose tissue, as well as in the gut, liver, and skin [36]. The proportion of ILC2s within the small intestine is higher in germ-free mice, likely due to the decrease in microbe-dependent populations. ILCs are also evolved with tolerance mechanisms regarding interactions between the host and the commensal microbiota. ILC3s are defined by the production of the Th17/22-associated cytokines IL-17 and/or IL-22, which enable them to promote immunity to extracellular bacteria and fungi, as well as tissue repair [37]. Recent studies have begun to disclose how ILC3s, a heterogeneous group found mainly in mucosal tissues, interact with gut bacteria, diet-derived factors, and various cell types to maintain intestinal homeostasis [36]. Even though a relationship between vitamin D and ILC2 has not been discovered yet, Chen et al. [38] demonstrated that vitamin D receptor knockout mice had more IL-22-producing innate lymphoid cells (ILC3) and more anti-bacterial peptides than wild type mice. Vitamin D also downregulates the IL-23 receptor pathway in human mucosal ILC3 [39].

### 4.2. Adaptive Immune System

#### 4.2.1. T Lymphocytes

Activation of T cell-mediated responses as part of an adaptive immune system is required for infection clearance. On the other hand, aberrant, overactive, or prolonged responses are also associated with chronic inflammatory conditions. Among other function, vitamin D favors T cell differentiation and function. CD4 T cells are considered to have high plasticity, being able to switch from subtypes and functional capabilities depending on the environmental triggers received; most of these environmental factors come from the microbiota and its metabolites, although the exact mechanisms and modulating compounds are still largely unknown [2]. Calcitriol and vitamin D-VDR signaling directly target T cells to stimulate the more tolerogenic Treg subpopulation in favor of inflammatory effector T (Teff; Th1, Th2, and Th17) cells [34,40]. T helper (Th) 17 cells and regulatory T cells (Treg) are antigen-specific populations that respond to transforming growth factor-β and retinoic acid and control immune tolerance. CD4^+^ Tregs, which express the transcription factor forkhead box P3 (Foxp3), are highly present in the gut lamina propria (LP), particularly in the colon, where they play a critical role in the maintenance of immune homeostasis [41]. Th17 cells participate in the pathogenesis of autoimmune and allergic diseases and take part in the host′s protective immunity, whereas Treg cells play an important role in the control of autoimmune reactivity [1]. The gut microbiome, too, plays an important role in immunomodulation since it promotes both humoral immunity (B cell development and proinflammatory T cell responses) as well as immune regulation (regulatory B and T cells) [6]. Moreover, the loss of a specific bacterial species can lead to overreaction or suppression of the innate immune response [42]. For instance, numerous studies have shown that the intestinal microbiota is closely associated with the balance of Th17 and Treg as they are significantly decreased in germ-free mice [1]. Segmented filamentous bacteria (SFB) strongly induce intestinal Th17 cells, which play a role in host resistance against intestinal pathogens and promote systemic autoimmunity. Among the indigenous commensal bacteria, Clostridium spp are great inducers of Tregs in the colon [41]. Candida albicans enhanced Th9 cell development in CD4 T cells residing in Peyer′s patches and mesenteric lymph nodes [2]. Bacteroides fragilis has an inhibitory effect on Th17 cells and has also been shown to stimulate IL-10-producing B cells (which are capable of suppressing T cell-mediated inflammation). Furthermore, co-infection of segmented filamentous bacteria and Listeria monocytogenes resulted in the production of Th17 and Th1 cells, demonstrating that individual bacteria can elicit specific immune cell responses [6]. Furthermore, it has recently been found that metabolites of the microbiota such as adenosine triphosphate (ATP) and SCFA stimulate the differentiation and development of Th17 and Treg cells, respectively. ATP, derived from the intestinal microbiota via P2X receptors, promotes the expression of proinflammatory cytokines that induce the development of Th17 cells and inhibit the production of Treg. ATP also modulates the functions of immune cells through P2X and P2Y receptors; these can activate a unique subset of CD70 high CD11 low cells that express different molecules that induce the differentiation of Th17 cells, such as IL-6, IL-23p19, and integrin-αV and TGFβ-activating -β8 [1]. A link between vitamin D, the microbiome, and regulatory T cells in the colon was proposed by Cantorna et al. [43]. In their study, they indeed discovered that vitamin D-sufficient (D+) mice had significantly higher frequencies of FoxP3+ and RORγt/FoxP3+ Treg cells in the colon compared to vitamin D-deficient (D–) mice. Th9 cells are a proinflammatory CD4 T cell subset characterized by a potent secretion of interleukin-9 (IL-9). They are involved in protection against parasitic infections [44] and anti-tumor immunity [45], but they are also linked to pathologies as described for allergic conditions, such as asthma, allergic rhinitis, atopic dermatitis (AD), and food allergies [46]. Even though the modulation of Th9 cell responses is not fully understood, these cells might be stimulated by microbial species such as *Staphylococcus aureus* and *Candida albicans*, whereas on the contrary, microbial and dietary compounds such as retinoic acid (RA), butyrate, and vitamin D show suppressive capacity on allergy-related Th9 responses [2].

#### 4.2.2. B Lymphocytes

In the intestines, B cells primarily localize to the lamina propria (LP). Gut microbiota is associated with and may potentially serve as an antigen source for immature B cell development in the gut. In fact, intestinal LP B cells have been found to express Rag2 and DNA polymerase, molecules characteristic of pro-B cells, suggesting that B cell development may occur in the intestine. This study also found that colonized mice showed significant increases in Rag1 and Rag2 and increased percentages of pro-B cells in the bone marrow, spleen, and lamina propria compared to germ-free mice [6]. The gut microbiome also promotes the differentiation of regulatory B cells (Breg, which represent about 10% of total circulating B cells) in the spleen as well as in the mesenteric lymph nodes. Bregs hamper inflammation-inhibiting Th1 cell activation, Th17 differentiation, and preservation of the Treg cell population [4]. The interaction between the microbiota and B cells also influences one another in maintaining homeostasis. In arthritis-induced mice, for example, the gut microbiome stimulated IL-1β and IL-6 production, which promoted the development and function of splenic and mesenteric lymph node IL-10-producing B cells [47]. The relationship between vitamin D and B lymphocyte has not yet been established, even if 1,25(OH)2D3 appears to regulate Breg cells and stimulate IL-10 production [4].

### 4.3. Other Interactions

#### 4.3.1. Microbiome Composition

Dysbiosis is an imbalance of commensal and pathogenic bacteria and the production of microbial antigens and metabolites. It is associated with several risk factors, such as the mode of delivery, formula feeding, the use of antibiotics, bacterial characteristics, mucosal cell characteristics, and the type of diet [48]. Dysbiosis has been linked to both the onset of intestinal inflammatory processes and extra-intestinal conditions associated with chronic inflammation and metabolic dysfunction, such as metabolic syndrome, obesity, and atopic disease [49]. Multiple studies have indeed shown that vitamin D deficiency or inactivating polymorphisms in VDR are associated with microbiome dysbiosis, with consequent increases in Bacterioides and Proteobacteria phyla and inflammatory disorders [6]. For example, vitamin D supplementation in Crohn’s disease patients promotes the growth of certain bacterial species in the gut microbiome, such as *Alistipes, Barnesiella, Roseburia, Anaerotruncus, and Subdoligranulum.* In patients affected by multiple sclerosis, it is associated with an abundance of *Akkermansia*, which promotes immune tolerance, as well as *Faecalibacterium* and *Coprococcus*, which both produce butyrate, an anti-inflammatory SCFA [50]. Moreover, Chatterjee et al. [51] demonstrated that VDR knockout mice have a different microbiome composition. Their fecal samples, as a matter of fact, contained *Clostridium, Eubacterium, Bacteroides, Tannerella*, and *Prevotella*, whereas control mice samples contained *Lactobacillus, Butyricimonas*, and *Lactococcus*. Vitamin D deficiency also has a key role in airway microbiome composition, as weekly oral supplementation has an impact on patients with cystic fibrosis [17]. In addition to this, vitamin D levels in utero (and shortly after birth) could also influence the microbial community that colonizes the infant gut, for example, an increased abundance of Lachnospiraceae/U. Clostridales and Lachnobacterium and a decreased abundance of Lactococcus in infants with higher cord blood vitamin D [10]. In childhood, gut microbiome composition is important to the development of food allergy, as atopic children’s microflora appears to be different from normal controls [52]. Children with a non-IgE-mediated cow’s milk allergy (CMA), for example, showed Bacteroides ad Alistipes abundance before dietary treatment. Bacteroides were even higher in children with an IgE-mediated cow’s milk allergy, suggesting a key role of these bacteria in the pathogenesis of CMA, as they have been reported to alter gut permeability [53]. Lower levels of vitamin D in early life, too, are more likely to lead to the development of food allergies, suggesting a possible link with microbiome development [26]. Unique microbial patterns have also been shown in autoimmune diseases such as inflammatory bowel disease, rheumatoid arthritis, multiple sclerosis, and systemic lupus erythematosus [6].

#### 4.3.2. Intestinal Barrier Integrity

The intestinal barrier is composed of a thick mucus covering the gut epithelium; the outer mucosal layer hosts commensal bacteria, the microbiome; and the inner layer, which is denser, excludes bacteria and external antigens, preventing their interaction with gut cells. Vitamin D/VDR signaling has an important role in maintaining epithelial integrity by regulating the expression of tight junctions (TJ) and adherent junctions (AJ). When this layer is compromised and the bacteria gain access to the lamina propria, vitamin D also has an impact on the activation an suppression of the innate and adaptive immune system at this site [33,54]. VDR signaling also maintains the integrity of the intestinal barrier by arresting inflammation-induced apoptosis of epithelial cells [55]. An increased permeability in the intestinal barrier has been associated with both intestinal (gastric ulcers, IBD, infectious diarrhea) and extraintestinal diseases (allergies, infections, chronic inflammations, type-1 diabetes, and behavioral disorders) [56,57]. In early life, changes in the development of the intestinal barrier is indeed linked to a greater exposure to luminal antigens and to accelerated immunological responses, resulting in clinical manifestation such as food allergies [58].

#### 4.3.3. Short-Chain Fatty Acids

Dietary and bacterial metabolites are known to influence immune responses, but they can also influence each other. For example, fiber intake is capable of making changes in the gut and airway microbiome by increasing species such as Bacteroidaceae and Bifidobacteriaceae. This leads to a local and systemic increase of circulating short-chain fatty acids (SCFAs) such as acetate, butyrate, and propionate, produced by bacterial fermentation of dietary fiber [59]. SCFAs bind to G protein-coupled receptors (such as GPR43, GPR41, and GPR109A), which play fundamental roles in the promotion of gut homeostasis and epithelial integrity and the regulation of inflammatory responses, and influence Treg biology, DC biology, and IgA antibody responses [60]. The principal anti-inflammatory effects of short-chain fatty acids are IL-12 inhibition; upregulation of IL-10 production in monocytes; repression of pro-inflammatory molecule release such as TNFα, IL-1, and NO; and reduction of NF-kB expression [61]. Butyrate and propionate, but not acetate, are known to promote Treg differentiation, thus influencing the balance between pro- and anti-inflammatory responses [62]. Additionally, butyrate leads to a downregulation of LPS-induced proinflammatory cytokine production (for example, IL-6, IL-12) by intestinal macrophages, the most abundant immune cell type in the lamina propria [63]. In addition, butyric acid is the preferred source of energy for colonocytes and controls the proliferation, differentiation, and apoptosis of colonocytes. By decreasing the intestinal epithelial permeability and increasing the expression of tight junction proteins and the production of mucin and antimicrobial peptides, it also strengthens the defensive barrier of the colon [64]. As reported before, vitamin D has a similar action of maintaining intestinal barrier integrity, suggesting a possible synergic effect with butyrate. Another link between SCFAs and vitamin D is the fact that they cooperate in raising host defense peptide (HDP) synthesis, a component of the innate immune system with antimicrobial and immunomodulatory activity [65]. Butyrate also increases VDR expression and the prodifferentiative effect of 1,25-dihydroxyvitamin D3. Furthermore, VDR participates in the butyrate-mediated inhibition of NFκB activation in human colon carcinoma cells [4]. SCFAs represent a strong connection between diet, microbiome, and allergic manifestations: Polymorphisms in GPR65, a G protein-coupled receptor highly expressed by mast cells and eosinophils and activated by protons in acid conditions, are linked to asthma and allergic diseases other than IBD [60]. To highlight the relationship between SCFAs and allergy, Canani et al. [53] also demonstrated that non-IgE-mediated CMA children presented a lower fecal concentration of butyrate compared to control, and that this could be related to an abundance of Lachnospira and Bacteroides in allergic children.

#### 4.3.4. Oxidative Stress

The imbalance between oxidants such as reactive oxygen species (ROS) and reactive nitrogen species (RNS) and antioxidants in favor of the first ones is known as oxidative stress, a condition that can lead to biological damage. ROS, such as hydrogen peroxide, hydroxyl radical, superoxide anion, and peroxynitrite, is derived from enzymatic reactions in various cell compartments, such as the cytoplasm, cell membrane, endoplasmic reticulum, mitochondria, and peroxisome. These are very unstable molecules containing oxygen reduced with added electrons; they can easily react with other molecules such as DNA, RNA, and proteins, playing a key role in the pathogenesis of various human diseases, including allergies [66]. Mitochondria release ROS in a controlled way as part of the normal metabolism; however, during stress or mitochondrial damage mitochondrial ROS (mtROS) can increase, promoting inflammation [67]. During health, for example, ROS are important for maintaining epithelial homeostasis. Intestinal epithelial cells protect themselves against pathogens by producing ROS, antimicrobial peptides, and mucus. This facilitates the mobilization of resident stem cells to replace lost cells and maintain epithelial integrity. mtROS also gain protection from colitis by activating the nuclear factor κB (NF-kb), which regulates a broad array of genes involved in inflammatory and immune responses [68]. At higher concentrations, ROS and other oxidants can also influence cell signaling pathways and molecules at cytoplasmic and nuclear levels. For example, the modulation of pattern recognition receptors (PRRs) such as Toll-like receptors (TLRs), a major contributor to inflammatory pathways, has recently been associated with mechanisms of oxidative biology. This, together with antigen exposure and further modulation of intrinsic signaling molecules such as NF-κB, is of absolute importance in devising future strategies for better therapeutic control of the clinical features of asthma [69]. Oxidative stress often correlates with dysbiosis by decreasing the microbial diversity in the gut and by promoting the outgrowth of specific bacteria through nitrate and tetrathionate respiration, such as *Salmonella* and *Citrobacter*. It is believed that oxidative stress induced by ROS increases the inflammatory reaction, leading, probably through a positive feedback mechanism, to an increased production of ROS and subsequent tissue damage [70]. Vitamin D regulates ROS levels through its anti-inflammatory effects and mitochondrial-based expression of antioxidants through cell-signaling pathways. Vitamin D, together with Klotho and nuclear factor-erythroid-2-related factor 2 (Nrf2), regulates the expression of many of the antioxidant systems that prevent oxidative stress by removing ROS and also by reversing the oxidative changes that occur during normal ROS signaling [71]. Klotho is an anti-aging gene whose expression is regulated by vitamin D. It mediates cellular signaling systems, including the formation of antioxidants. Nrf2 is a transcription factor that translocates to the nucleus following calcitriol–VDR interaction. It activates the expression of several genes that have antioxidant activity. When Nrf2 activity is insufficient, risks from oxidative stress-related tissue damage increases, resulting in excessive ROS formation and leading to a pathologic oxidative stress cycle [72].

## 5. Focus on Specific Allergies

### 5.1. Asthma

Asthma is a respiratory condition that likely results from complex interactions between multiple environmental and genetic influences. Proposed risk factors for asthma vary with the age of asthma onset and timing of exposures and behaviors relative to the onset of asthma. Microbes have long been postulated to play a role in asthma and might also shape its heterogeneity [73]. The early life microbiome likely influences the likelihood that an allergic predisposition results in asthma. Exposure to bacteria and bacterial products may influence the development of allergen sensitization and asthma, although the exact effects appear to depend on a complex interplay of timing of exposure (first year of life versus later in life), location, abundance, and diversity of the microbiome, and specific microbial products [74]. As an example, early life exposure to allergens and certain bacteria in the environment may lower the risk of asthma [75], whereas later life exposure to bacteria may increase the risk of asthma [76]. The mechanism for this protective effect is not known, but changes in gut microflora and related effects on innate immunity are hypothesized. Moreover, studies have suggested that interactions between RSV and nasopharyngeal microbiota might modulate the host immune response, potentially affecting clinical disease severity such as the risk of wheezing and symptoms of asthma. Respiratory virus infection has been shown to alter the airway microbiome, with increased detection of *Streptococcus pneumoniae*, *Moraxella catarrhalis*, and *Haemophilus influenzae* in children with and without asthma [77]. This coalescing of antimicrobial immune responses was associated with increased respiratory symptoms. Since most asthma has its origins in childhood, early nutrition, including prenatal exposure to nutrients, may be relevant as a risk factor for the development of asthma and allergies [78]. It remains unclear whether vitamin D supplementation has a role in asthma prevention. There are conflicting reports on the association between vitamin D status and allergic diseases. In some reports, vitamin D deficiency (in pregnant women, children, or adolescents) was associated with increased as well as with decreased frequency of allergic diseases such as asthma or eczema [79,80]. The Vitamin D Antenatal Asthma Reduction Trial (VDAART) attempted to answer the question of whether vitamin D supplementation to women during pregnancy can prevent the development of asthma and allergies in their children. The aim of this study was to determine whether children born to mothers who had received 4400 IU of vitamin D3 per day during pregnancy (vitamin D group) would have a lower incidence of asthma and recurrent wheeze at the age of six years than would those born to mothers who had received 400 IU of vitamin D3 per day (control group). The six-year follow up showed that the prenatal period alone did not influence the incidence of asthma and recurrent wheeze among children who were at risk for asthma [81,82]. Randomized trials examining the effect of vitamin D supplementation on asthma outcomes were inconclusive. Larger trials with longer-term follow-up are needed.

### 5.2. Atopic Dermatitis

A multiplicity of factors, including skin barrier abnormalities, defects in innate immunity response, Th2-skewed adaptive immune response, and altered skin resident microbial flora are involved in the pathogenesis of atopic dermatitis [83]. The increased prevalence of AD, particularly in industrialized regions, has been hypothesized to be due to excessive hygiene accompanying the Western lifestyle, reducing exposure of the host’s immune system to education provided by beneficial microbes. A subset of patients with AD are prone to microbial dysbiosis with bacteria, viruses, and fungi, which can exacerbate skin inflammation [84]. For example, *Staphylococcus aureus* is the most common skin-cultured pathogen of AD and its colonization is associated with increased IgE responses, food allergy, and AD skin-disease severity [85,86]. There is increasing evidence that commensal skin microbes from normal skin can improve the skin barrier and augment host defense against skin pathogens, including *S. aureus*. *Staphylococcus epidermidis* and *Staphylococcus hominis* secrete antimicrobial activities that inhibit S. aureus growth and biofilm formation [87,88]. *S. epidermidis* has also been found to stimulate Toll-like receptor 2 to induce production of keratinocyte-derived antimicrobial peptides and increase tight junctions to enhance the skin barrier [89]. These protective strains are deficient in atopic dermatitis [85]. Using antibiotics in the treatment of *S. aureus* infection is disadvantageous because they kill not only *S. aureus*, but also beneficial bacteria and potentially select antibiotic-resistant bacteria such as methicillin-resistant *S. aureus*, underlining the importance of maintaining the skin-resident normal microbial flora. A few small, randomized trials have evaluated the role of vitamin D supplementation in the prevention of winter-related exacerbation of atopic dermatitis [90,91,92]. In the largest study, 107 children with a history of atopic dermatitis worsening during winter were treated daily with 1000 international units of vitamin D or placebo for one month. The primary outcome was a reduction in the clinician-measured eczema area and severity index (EASI). At the end of the study, the mean decrease in the EASI score was 6.5 in the vitamin D group and 3.3 in the placebo group [90]. Although the results of these trials suggest that winter supplementation of vitamin D may be beneficial for patients with atopic dermatitis, larger, well-designed studies are needed to clarify the role of vitamin D in the prevention and treatment of atopic dermatitis.

### 5.3. Food Allergy

Tolerance is the normal immune response to the food an individual eats over a lifetime [93]. Food allergy is thought to involve deviation from the default state of mucosal immune tolerance that can be driven by diet, commensal microbiota, and the interactions between them [94]. Although alterations in the intestinal microbiome are known to be associated with the development of asthma, less is known regarding the role of microbiome alterations in food allergy development. In their prospective microbiome-wide association study of food sensitization and food allergy in early childhood, Savage et al. [95] collected intestinal microbiome samples at age three–six months in children participating in the follow-up phase of VDAART. As a result, the study suggested that the microbiome may have a causal role in the development of food allergy. Specific patterns of microbiota colonization, such as colonization in large numbers by probiotics or greater microbial diversity, may favor tolerance, possibly through increased production of IgA and IL-10 [96]. It is postulated that cesarean delivery compared to natural childbirth increases the risk of IgE-mediated sensitization to food allergens as a result of alterations in the gut microbiota [97]. Data on the impact of cesarean delivery on the rate of clinical food allergy are inconsistent, although most studies show an increased risk [98,99]. Studies of children with a milk allergy have shown that infants with a milk allergy have higher total bacteria and anaerobic counts compared to healthy control subjects [100]. Reduced microbial richness and increased Bacteroides were observed in subjects with a self-reported peanut or tree nut allergy compared to those not reporting these allergies [101]. Murine models of food allergy have provided experimental dimensions to the study of the microbiota and food allergy. In a mouse model of induced food allergy by skin sensitization, feeding mice a high-fat diet prior to skin sensitization and subsequent gastrointestinal challenge to a food allergen was associated with the development of obesity, decreased intestinal bacterial diversity, and increased food allergy susceptibility [102]. Transfer of the gut microbiome from these mice to germ-free recipient mice on a normal diet led to decreased intestinal bacterial diversity and increased food allergy susceptibility, but not obesity. Translation of these observations to humans requires further study. Some murine studies have shown that Clostridia strains in particular modulate allergy. Imbalances in T helper cell subsets (Th1/Th2) in cellular immunity contribute to the development of food allergy. Seventeen strains of bacteria from human stool that enhance regulatory T-cell abundance have been identified. Oral administration of these 17 Clostridia strains into mice attenuated disease in models of colitis and allergic diarrhea [103]. Therapy with a consortium of protective clostridial species suppresses food allergy in mice [104], suggesting that gut microbiota dysbiosis is a potential target for future treatment of food allergy if these findings are replicated in humans. Similarly, skin commensal bacteria have been recognized as significant factors imprinting the immune system [105], possibly also influencing food allergy. Skin colonization with *S. aureus*, a marker for more severe eczema, is also associated with sensitization to food allergens [85]. Independent of eczema severity, skin *S. aureus* colonization was associated with hen′s egg and peanut sensitization and persistent allergy, with a weaker association seen for cow′s milk, in a study that followed children from infancy to six years of age [106]. As a matter of fact, although strict allergen avoidance is still the key treatment for food allergy, there is a greater focus on the preventive effect of the addition of lactose and probiotics to hypoallergenic infant formulas in order to modulate the gut microbiome and early immune responses in high-risk populations, either in families with a history of allergies, or in infants who are showing evidence of food sensitization or eczema [28,107]. Only a few trials of probiotics for the prevention or treatment of challenge-proved food allergies have been published. Trials of probiotic supplementation with *Lactobacillus casei* and *Bifidobacterium lactis* for 12 months showed no effect on milk allergy resolution, although *Lactobacillus rhamnosus* combined with extensively hydrolyzed casein formula increased rates of milk allergy resolution compared to a control group receiving formula alone [108,109]. The probiotic Lactobacillus rhamnosus GG administered with peanut oral immunotherapy for 18 months induced desensitization compared to a placebo. However, because there was no oral immunotherapy-only or probiotic-only group, the efficacy of the probiotic itself is unclear. The effects of probiotic treatment are likely strain specific, and the data are currently inconclusive to support probiotic supplementation with specific taxa for food allergy [110]. Early infancy could be the key window of opportunity for intervention given age-dependent associations between the gut microbiome and food allergy outcomes. Gut microbial richness at the age of three months is associated with an increased likelihood of food sensitization by the age of one year [111]. Murine models also support age-sensitive interactions with microbiota [112]. Colonization of germ-free mice with a diverse microbial population early but not late in life suppresses IgE and prevents mice from having a food allergy [113]. These collective findings support the notion that microbial effects on early immune system development play a role in subsequent food allergy development. Epidemiologic studies suggest possible associations between vitamin D deficiency and a variety of conditions, but a causal relationship has not been established. These include certain immunologic conditions such as food allergies and asthma in adolescents [114]. Vitamin D has a potentially significant role for improving the symptoms and severity of food allergy for its immunomodulatory effects. Proposed but unproven theories suggest a role for both vitamin D excess and deficiency in the development of food allergy. Future studies for assessing adequate amounts of vitamin D intake for the treatment and prevention of food allergy are also warranted.

## 6. The Controversial Use of Probiotics in the Prevention and Treatment of Allergies

Probiotics are defined as live nonpathogenic microorganisms, which, when consumed in adequate amounts, confer a health effect on the host. The most frequently used bacterial genera are lactic bacteria (Lactobacillus and Bifidobacterium genera). The mechanisms of action of probiotics in humans are not yet fully known, as they are mainly documented through in vitro or animal model studies [115]. To date, there are contradictory results among studies about the use of probiotics in the prevention and/or treatment of allergies. Although the European Academy of Allergy and Clinical Immunology (EAACI) [116] and the Nutrition Committee of the European Society of Paediatric Gastroenterology, Hepatology and Nutrition (ESPGHAN) [117] agreed that there is too much uncertainty to draw reliable conclusions from the available data and that there is no evidence to recommend prebiotics or probiotics or other dietary supplements based on particular nutrients to prevent food allergy, the World Allergy Organization (WAO) [118] suggested considering the use of probiotics in (a) women pregnant with children with a high risk for allergy, (b) women who breastfeed infants at high risk of developing allergy, and (c) infants at risk of developing allergies, because there is a net benefit resulting in the primary prevention of eczema. The necessity of further in vivo studies to investigate once and for all the use of probiotics is evident.

## 7. Conclusions

In this review, we discussed an abundance of evidence that demonstrates the importance of the relationship between the microbiome and vitamin D. The microbiome and vitamin D deeply influence each other and the immune system in many different ways. It is evident that the immune system and the microbiome are interconnected, and that vitamin D is a critical intermediary player in this dynamic. For example, alterations in vitamin D/VDR signaling are associated with microbiome dysbiosis, which in turn has been linked to both intestinal inflammatory processes and extra-intestinal conditions such as atopic disease. From this point of view, the daily intake of a sufficient amount of vitamin D (400 IU vitamin D/day for children <1 year old, 600 IU vitamin D/day for people aged 1–70 years old, and 800 IU vitamin D/day for people >70 years old) or its oral implementation in the case of a deficit, seems fundamental to prevent the development of allergies from infancy. On the other hand, a correct microbiome composition can also be preserved by the use of probiotics when commensal microorganisms are consumed (during antibiotic treatment or certain pathologies such as colitis). In conclusion, considering the excellent safety and economic nature of vitamin D and probiotics, it appears plausible that vitamin D and probiotic supplementation can help prevent the development of infections and regulate the immune response by preventing the development of allergies.

## Figures and Tables

**Figure 1 ijms-22-04288-f001:**
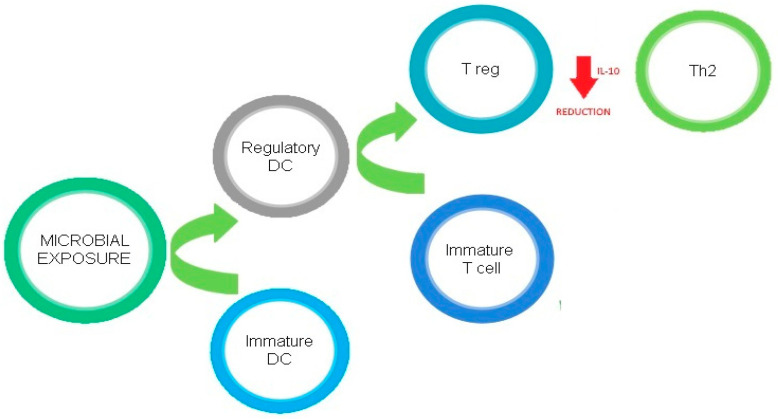
Regulatory effects of microbial exposure on Th2 cells.

**Figure 2 ijms-22-04288-f002:**
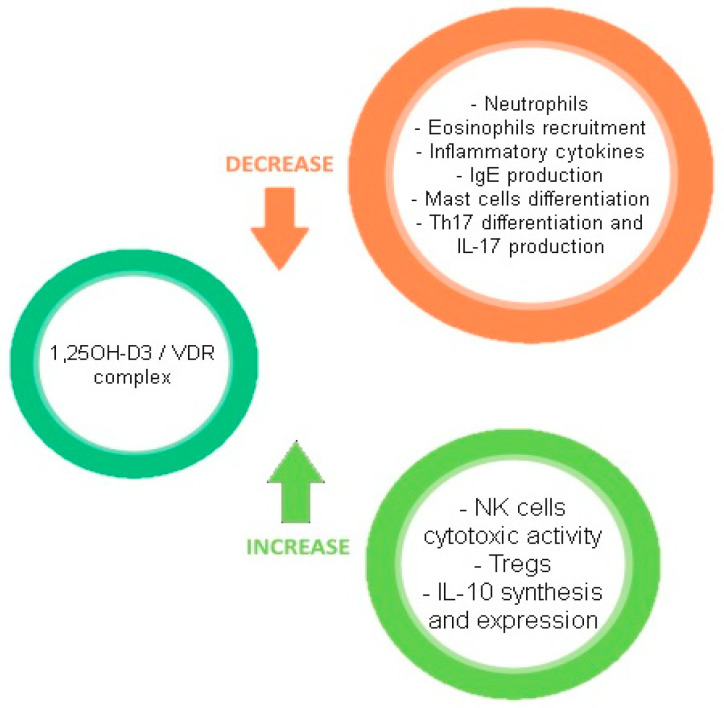
Effects of the VD/VDR complex on the immune system.

**Figure 3 ijms-22-04288-f003:**
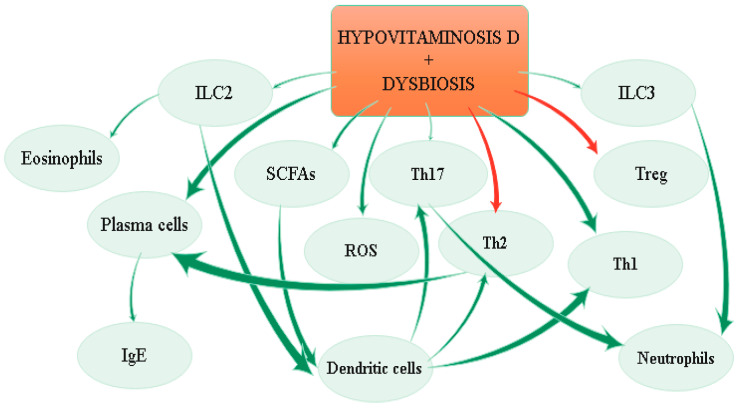
Effects of hypovitaminosis D and dysbiosis on the immune system.

## Data Availability

The study does not report any data as a review article and not a research article.

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
