# Peer review of "Vitamin D and Microbiota: Is There a Link with Allergies?"

_ijms, 2021, doi:10.3390/ijms22084288_

Round 1

Reviewer 1 Report

I appreciate all the hard work the study group invested in this review. I sincerely hope my comments will be taken as being constructive and will become useful for a revision. This review was supposed to shed a new light on interaction of microbiota with vitamin D and implicitly the immune system - and possible implications for allergies. The topic is a hot one nowadays. However, this review did not bring anything new. Instead, there is a plethora of sentences about other diseases, so that the reader may not see the forest for the trees. There have been many excellent published reviews focusing on vitamin D and diseases, microbiota etc. Please focus on the topic – allergies and remove all unnecessary sentences. Please organize better your material, in order to really benefit readers and their patients. Also, since the probiotics are discussed, maybe they should be included in the title. Not that they really help, according to the most recent guidelines. The authors should pay more attention to details and insert more figures, which would really help. Otherwise, I cannot recommend this manuscript to be published, since there is nothing new.

Specific comments:

  • 2. Airways microbiota: please insert reference “Santacroce L, Charitos IA, Ballini A, Inchingolo F, Luperto P, De Nitto E, Topi S. The Human Respiratory System and its Microbiome at a Glimpse. Biology (Basel). 2020 Oct 1;9(10):318.”
  • 4 Page 4 – lines 132-135 “The relationship between the microbiota and neoplastic development was also studied: in patients with small cells lung carcinoma, the anti-cancer effects of probiotics was confirmed by the downregulation of miR-150 and the over-expression of miR-143..” There is no reference added – please insert the correct one (however, this is not about allergies). Any sentence taken from any published paper should be referenced. In this case: “RODRÍGUEZ-NOGALES A, ALGIERI F, GARRIDO-MESA J, VEZZA T, UTRILLA MP, CHUECA N, GARCIA F, OLIVARES M, RODRÍGUEZ-CABEZAS ME, GÁLVEZ J. Differential intestinal anti-inflammatory effects of Lactobacillus fermentum and Lactobacillus salivarius in DSS mouse colitis: impact on microRNAs expression and microbiota composition. Mol Nutr Food Res 2017; 61: 1700144-1700156”. Page 4 – Lines 136 – 138 – for the sentence “A recent research showed also that lung cancer microbiota is enriched in Proteobacteria…”, again the reference is not inserted and not listed. Please insert the correct one: “Gomes, S., Cavadas, B., Ferreira, J.C. et al. Profiling of lung microbiota discloses differences in adenocarcinoma and squamous cell carcinoma. Sci Rep 9, 12838 (2019).” (However, this refers to lung cancer and not allergies). This paragraph is not appropriately organized – please focus on the topic…Please delete sentences about CRC – no interest for this topic.
  • 2. and 3.3 and 3.4 and the other following paragraphs: Again, this review should focus on vitamin D and allergies. There are many unnecessary sentences in these paragraphs, which are well known and not related to the topic. Please remove them.
  • Figure 1 is very scarce; it should be adequately designed. More figures should be added (from paragraph 4 - Linking Vitamin D, Microbiome And Immune System and paragraph 5), as they would be really helpful for readers. Please also focus, in the main text, only on allergies (sentences about infections, IBD, autoimmune disorders, cancers etc do not have any place here, as this is not the focus of this review).
  • There is “4.2. Adaptive immunite system” (should be immunity), but 4.1 is not inserted.
  • Even if in the reference list there are titles including Covid-19 and vitamin D, the main text did not include anything about this. Please use the findings of these references in the main text. If not, please just delete those references.
  • Reference 10: Please delete “a” in Sordilloa”.
  • P1 line 33: please replace “as a matter of fact” – repetition with previous sentence.
  • References are not written in a uniform style: some start with family name, some with given name. Please correct.
  • Some references that could be added:

Mirzakhani H, Al-Garawi A, Weiss ST, Litonjua AA. Vitamin D and the development of allergic disease: how important is it? Clin Exp Allergy. 2015 Jan;45(1):114-25

Sikorska-Szaflik H, Sozańska B. The Role of Vitamin D in Respiratory Allergies Prevention. Why the Effect Is so Difficult to Disentangle? Nutrients. 2020 Jun 17;12(6):1801

Savage JH, Lee-Sarwar KA, Sordillo J, Bunyavanich S, Zhou Y, O'Connor G, Sandel M, Bacharier LB, Zeiger R, Sodergren E, Weinstock GM, Gold DR, Weiss ST, Litonjua AA. A prospective microbiome-wide association study of food sensitization and food allergy in early childhood. Allergy. 2018 Jan;73(1):145-152.

Koplin JJ, Peters RL, Allen KJ. Prevention of Food Allergies. Immunol Allergy Clin North Am. 2018 Feb;38(1):1-11.

Polk BI, Bacharier LB. Potential Strategies and Targets for the Prevention of Pediatric Asthma. Immunol Allergy Clin North Am. 2019 May;39(2):151-162.

Sardecka I, Krogulska A, Toporowska-Kowalska E. The influence of dietary immunomodulatory factors on development of food allergy in children. Postepy Dermatol Alergol. 2017 Apr;34(2):89-96.

Tuchinda, P., Kulthanan, K., Chularojanamontri, L. et al. Relationship between vitamin D and chronic spontaneous urticaria: a systematic review. Clin Transl Allergy 8, 51 (2018)

Author Response

Dear Editor, I submit the revised version of our paper Vitamin D and microbiota: is there a link with allergies? I have attached as word file the response to referees.

Giuseppe Murdaca

Reviewer 2 Report

The authors summarize the influence of the microbiome and vitamin D on the immune system with a particular focus on allergic diseases and discuss the necessity of further studies on the use of probiotics  and of a correct intake of vitamin D.

COMMENTS

In my opinion the paper is well constructed. It summarized the state of  current knowledge including sostantive findings, as well as theoretical and methodological contributions to this particular topic.

Could  you  resume the best hypotesis about   how Vitamin D deficiency could link with allergies?  

In your opinion Vitamin D deficiency  is  cause or effect of dysbiosis?

MINOR REMARKS

At line 195 please add l  ad m = ml

At line 299 Treg as instead  Tre gas

Author Response

Dear Editor, I have attached the response to referees as word file.

Giuseppe Murdaca

Round 2

Reviewer 1 Report

Generally, the authors made the changes suggested by the reviewers. The manuscript appears much better now.

Other than that, just a minor comment: Please insert 119 – for the last reference: Schünemann, H. J. World Allergy Organization-McMaster University Guidelines for Allergic Disease Prevention (GLAD-P): Probiotics. World Allergy Organ. J. 2015, 8 (1), 1–13. https://doi.org/10.1186/s40413-015-0055-2.

Author Response

Dear Editor many thank for the referee's suggestion. However, the reference that the referee 1 suggested to insert in the text as reference 119 has been yet considered and it is the reference 118. Furthermore, I have insert the figures in the text in the appropriate part. Regards Giuseppe Murdaca